# Palladium Supported on Porous Chitosan–Graphene Oxide Aerogels as Highly Efficient Catalysts for Hydrogen Generation from Formate

**DOI:** 10.3390/molecules24183290

**Published:** 2019-09-10

**Authors:** Aicha Anouar, Nadia Katir, Abdelkrim El Kadib, Ana Primo, Hermenegildo García

**Affiliations:** 1Instituto de Tecnología Química, Consejo Superior de Investigaciones Científicas, Universitat Politècnica de València, Av. De los Naranjos s/n, 46022 Valencia, Spain; 2Euromed Research Center, Engineering Division, Euro-Med University of Fès (UEMF), Route de Meknes, Rond-point de Bensouda, 30070 Fès, Morocco (N.K.) (A.E.K.)

**Keywords:** liquid hydrogen carriers, formate as hydrogen carrier, catalyst for hydrogen generation, palladium as catalyst for hydrogen generation, chitosan–graphene oxide as catalyst support

## Abstract

Adsorption of Pd(NH_3_)_4_^2+^ in preformed chitosan–graphene oxide (CS-GO) beads and their subsequent reduction with NaBH_4_ afford well-dispersed, high dispersion (~21%) of uniformly sized Pd nanoparticles (~1.7 nm). The resulting Pd/CS-GO exhibits interesting catalytic activity for hydrogen generation by ammonium formate decomposition. The optimal GO proportion of 7 wt% allows reaching, at 60 °C, a turnover frequency above 2200 h^−1^—being outstanding among the highest values reported for this process to date. Interestingly, no formation of CO or CH_4_ was detected. The catalyst did not leach, although it underwent gradual deactivation, probably caused by the increase in the Pd average size that became over 3 nm after three uses. Our results are relevant in the context of efficient on-board hydrogen generation from liquid organic hydrogen carriers in transportation.

## 1. Introduction

In the context of the on-going shift from fossil fuels to renewable energy sources, one of the viable possibilities in transportation is the use of hydrogen as fuel. Hydrogen will act as an energy vector and it will be obtained from water by electrolysis using renewable electricity or by any other primary energy source [1].

One of the main problems associated with the use of hydrogen as a transportation fuel is the storage under ambient conditions of sufficient amounts to reach the required automotive autonomy [2]. Among the various alternatives being considered, one is the use of liquid organic hydrogen carriers (LOHCs) that, on demand, generate on board hydrogen that can be used in a fuel cell. In this context, catalysis for hydrogen generation from LOHCs is one of the key components of the process [3,4]. Among the requirements that have to meet catalysts for hydrogen generation from LOHC, the most important ones are the high rates (turnover frequency) that have to be attained under moderate conditions. For this reason, development of catalysts for hydrogen evolution from different LOHCs has become a research front in the area of using hydrogen as fuel.

One of the most widely studied LOHC’s is formic acid and its salts. The main reasons for this preference is the non-gaseous physical state of formic acid and its derivatives, and their wide availability via the catalytic hydrogenation of CO_2_. A large number of studies describe various catalysts for formic acid and formate decomposition into hydrogen and CO_2_ [5]. Most of the catalysts for formate decomposition are transition metals having well established hydrogenating/dehydrogenating activity [6]. Among them, supported palladium exhibits a general activity, frequently higher than those of other metals [7,8,9]. These precedents have established that the catalytic activity of palladium nanoparticles (Pd NPs) is largely determined by the nature of the support on which they have been deposited. Typical supporters of Pd NPs for hydrogen evolution from formate decomposition are large surface area metal oxides such as TiO_2_, Al_2_O_3_, and active carbons. 

In the present manuscript, we investigate the catalytic activity of Pd NPs embedded on porous chitosan (CS) and chitosan–graphene oxide (CS-GO) beads. Chitosan is a natural, widely-available polysaccharide obtained by chitin deacetylation, the most important waste from the fishery industry [10]. Highly porous beads of chitosan with large surface area can be obtained by appropriate coagulation, gel handling, and supercritical drying. Among the possible applications of porous chitosan, its use as support of metal NPs, has been the subject of several studies [11,12]. In this context, we have previously demonstrated the efficiency of chitosan microspheres as a medium for the growth of Au, Pd, and Cu complexes to promote several organic transformations including Sonogashira coupling, C–S coupling, click cycloaddition, alcohol oxidation, etc. [13,14,15,16,17]. Alternatively, Pd NPs embedded in alginate beads have also been used as catalyst for Suzuki cross-coupling [18]. The presence of well-dispersed amino groups in the biopolymer backbone provides a way for stabilizing the growing metal clusters while the open-macroporous network allows for fast molecular traffic during catalysis. 

In spite of its advantages in terms of sustainability, large surface area, and the presence of amino functional groups for interaction with the metal NPs, one of the major drawbacks of porous chitosan as support in catalysis is the lacking mechanical stability of the beads under reaction conditions [19]. 

There have been several ways to increase the mechanical stability of these porous chitosan microspheres by crosslinking with glutaraldehyde, glyoxal, and other dialdehydes [20,21]. Another alternative that we used in the present study, was to use highly reactive graphene oxide (GO) sheets to crosslink chitosan fibrils [22]. GO has epoxide, hydroxyl, and carboxylic acid groups that can react with the chitosan primary amino groups forming covalent bonds that crosslink chitosan fibrils. In this way, the network constituted by the composite of chitosan and graphene oxide is more robust and can stand a wider range of temperatures and reaction conditions [22]. As it will be shown later, the entrapment of an appropriate proportion of GO in chitosan embedding Pd NPs as active sites provides a highly active catalyst for hydrogen generation at moderate temperatures from formate, reaching turnover frequencies that are among the highest reported so far for this reaction.

Pd NPs of very small size (~1.7 nm) were prepared embedded with CS-GO composite as submillimetric spherical beads. The materials were very active to promote ammonium formate decomposition to H_2_, with no formation of CO or methane as side products. Ammonium formate was selected in the present study due to its solubility, but alkali formates can also be used. Under optimal reaction conditions: methanol as solvent, 60 °C, and 7 wt% of GO in the support, Turn Over Frequency (TOF) values over 2200 h^−1^ were measured. This value compared favorably to precedents reported in the literature. The catalyst can be reused with gradual decay in its activity, probably due to agglomeration of Pd NPs that grow up to an average diameter of 3.67 nm in the fourth run. Considering the natural origin of chitosan, the present system represents a step forward towards sustainability of a process that could be the basis of novel auto motion fuels. 

## 2. Results

First, catalyst preparation and characterization is described before presenting their catalytic activity. Four samples having a similar Pd content (based on ICP/OES analysis) and differing in the GO content were prepared as illustrated in Scheme 1. The list of catalysts and their main analytical and physicochemical properties are summarized in Table 1. Briefly, acid chitosan solutions were coagulated by concentrated NaOH to form CS beads of about millimetric size. For those samples containing GO, a suspension of the appropriate amount of GO was first prepared by sonication and, then, the corresponding amount of chitosan in acid solution was added before gellification under basic conditions. The CS and CS-GO microspheres were washed with water and the resulting hydrogels were impregnated with Pd(NH_3_)_4_Cl_2_ solution. Subsequently, the Pd(NH_3_)_4_^2+^/CS and Pd(NH_3_)_4_^2+^/CS-GO hydrogels were converted into alcogels by dispersing them into a water/ethanol mixtures of increasing ethanol proportion. Pd NPs adsorbed on CS or CS-GO were formed by Pd(NH_3_)_4_^2+^ reduction with NaBH_4_ in ethanol. An instantaneous change in the color of the beads upon contacting NaBH_4_ was visually observed. Finally, the catalyst samples were subjected to supercritical CO_2_ drying to obtain macroporous aerogels of Pd/CS or Pd/CS-GO ready to be used as catalysts. 

Specific surface area and porosity were measured by isothermal N_2_ absorption at 77 K. In accordance with the literature [18], the preparation procedure, particularly supercritical CO_2_ drying, renders a Pd/CS sample with a large surface area value of ~220 m^2^/g. The presence of GO further increased the specific surface area of the material up to 430 m^2^/g, a factor that should be beneficial from the catalytic point of view. Furthermore, pore volume increased upon incorporation of GO in the composite from 1.06 to 1.51 cm^3^/g. The values of surface area and porosity are collected in Table 1.

After preparation, the Pd content of the different Pd/CS and Pd/CS-GO samples was analyzed and the results are also included in Table 1. The samples and the supports were characterized by FESEM. These images show CS beads of a large porosity in accordance with previous studies that have shown that the conversion of CS hydrogels to alcogels before supercritical CO_2_ drying afford samples with a large macroporosity and surface area [23]. FESEM images reveal a change in the morphology of the pores due to the presence of GO. These changes appear more clearly as the GO content increases. To illustrate the influence of GO on the porosity of the microspheres, Figure 1 compares two representative FESEM images where the presence of GO sheets filling some of the pores present in CS can be seen. In none of the cases was the presence of Pd NPs detected by FESEM. However, elemental analysis in FESEM showed the presence of Pd uniformly distributed throughout the particle (Figure 1c). Supporting information provides a set of images with elemental mapping for C, O, N, and Pd for the samples under study showing uniform Pd distribution. 

Dark-field STEM (DF-STEM) images clearly reveal the presence of small Pd NPs of nanometric and subnanometric particle size uniformly distributed throughout the catalyst beads, and allowed us to determine particle size distribution (Figure 2). The identity of the NPs as corresponding to Pd was confirmed by EDX analysis of selected areas (Figure 1). Table 1 includes the average particle size and standard deviation determined by counting a statistically relevant number of particles in these DF-STEM images. As it can be seen there, the samples under study exhibit a similar average size about 1.7 nm.

X-Ray Photoelectron Spectroscopy (XPS) analysis of the Pd/CS-GO2 sample showed the corresponding Pd peak appearing at a binding energy about 337 eV and 342 eV for 3d_5/2_ and 3d_3/2_, respectively. Deconvolution of the experimental XPS peak indicated that the main component corresponds to Pd(0) at 336.05 eV binding energy with an atomic proportion of 0.208%, accompanied by another component at 338.21 eV due to PdO in a proportion of 0.154% (Figure 3). Thus, the Pd(0)/Pd^2+^ ratio of the exposed, catalytically relevant layers of the Pd NPs in the fresh catalyst was 1.35.

### Catalytic Activity 

As mentioned in the introduction, the purpose of the present work was to develop a heterogeneous catalyst for hydrogen generation by decomposition of ammonium formate at moderate temperatures. All the catalysts tested were able to generate hydrogen from ammonium formate solutions. Initial studies were performed using Pd/CS in aqueous solution. It was found that hydrogen generation rate and final production increase with temperature ranging from 25 to 60 °C. Figure 4 presents the temporal profiles of hydrogen evolution at three different temperatures. The apparent activation energy (Ea) estimated from the Arrhenius plot was 50.35 kJ·mol^−1^.

According to the stoichiometry indicated in Equation (1), NH_4_HCO_2_ decomposition to H_2_ should give rise simultaneously to stoichiometric amounts of CO_2_. CO_2_ formation was also quantified by gas chromatography, observing a growth in the concentration of CO_2_ in the gas phase during the reaction. However, CO_2_ quantification always showed lesser amounts than expected based on the stoichiometric formation with H_2_. It is proposed that the reason for the lower-than-expected CO_2_ evolution is the solubility of CO_2_ in the liquid phase at neutral or basic pH values. In fact, the pH of the aqueous solution during Pd catalysed ammonium formate decomposition grows from an initial value of 7 to a final pH value of 9. This increase in the pH value is in agreement with the expected hydrolysis of NH_4_^+^ evolving some NH_3_.
NH_4_^+^ HCOO^−^ → H_2_ + CO_2_ + NH_3_(1)

A comparison of the catalytic activity of Pd/CS with a commercially available catalyst of Pd supported on activated carbon (10% Pd/AC) using the same total Pd content and identical reaction conditions showed the higher activity of Pd/CS compared to 10% Pd/AC (see Figure 5).

Stability of the Pd/CS catalyst was studied by performing consecutive reuses of the same sample. It was observed that both, initial reaction rate and final hydrogen production at 24 h, decreased upon reuse (see Figure 6). The electron microscopy study revealed that the average Pd NP size increased from a value of 1.7 nm for the fresh sample to 3.67–6.17 nm after the reaction. Supporting information (Figure 7) provides a set DF-STEM images of the used catalyst and the corresponding histogram of Pd particle size from which the average was determined. This particle size growth observed in Transmission Electron Microscopy (TEM) images was apparently the main reason responsible for the observed gradual deactivation of the catalyst. It has been proposed based on Density Functional Theory (DFT) calculations that the optimal particle size to achieve the highest activity in Pd NPs for H_2_ generation should be about 2 nm, and that the catalytic activity should decrease for larger particles.

In order to increase the stability of CS-supported Pd NPs and considering that aggregation of small size Pd NPs appears to be the main reason for deactivation, it was speculated that an increase in the mechanical robustness of CS should be beneficial by favouring the interaction of Pd NPs with the NH_2_ functional groups of CS that should result in a better immobilization of Pd NPs. In this regard, and as previously commented in the Introduction, it has been shown that GO in low percentages can act as crosslinker of chitosan fibrils enhancing the mechanical stability of the material that maintains the morphology of the catalyst beads under harsher conditions [24,25]. To test this hypothesis, three additional Pd-containing samples supported on CS were prepared using GO as a crosslinker.

Moreover, it has been previously found that some Pd catalysts may exhibit higher activity in methanol than in water [26]. For this reason, we also tested the activity of the Pd/CS-GO catalysts for H_2_ generation in methanol versus water as solvent. Remarkably, the catalytic activity for H_2_ generation at 25 °C from NH_4_HCO_2_ in methanol solution increased with the GO content of the support ranging from 3 to 12%. Figure 8 provides time–H_2_ evolution plots for the samples having different GO in the CS-GO beads. According to the literature [27], the beneficial influence of GO should be related to the mechanical stability of the CS-GO supports, maintaining surface area and porosity, thereby having a better metal-support interaction with the Pd NPs. In addition, the catalytic activity of the most active Pd/CS-GO2 sample was higher in methanol than in water, either at 25 or 60 °C. Figure 9 provides a comparison of the time–hydrogen generation plots in water and methanol.

Chemical analysis of methanol after removal of the most active Pd/CS-GO2 solid catalyst at 60 °C or below showed that the Pd leached from the solid catalyst to the solution was a negligible percentage (<0.006%) of the initial total Pd content present in the catalyst. In addition, a hot filtration test carried out at 60 °C after 1 h reaction showed that H_2_ does not evolve after removal of the solid catalyst, thus ruling out that the very minor Pd percentage leached from the solid could contribute in a measurable percentage to the H_2_ generation observed in the presence of the solid catalyst.

The results presented in Figure 6 for Pd/CS-GO2 as catalyst are also in agreement with the expected influence of the reaction temperature on H_2_ generation. The reaction rate and final H_2_ production were significantly higher at 60 °C (see Figure 9b). To calculate the TOF value, the amount of active sites was estimated using a dispersion value for Pd determined by CO chemisorption of 21.14%. Under the best working conditions for methanol at 60 °C using the most active Pd/GO-CS2, a turnover frequency at 60 °C of 2279 h^−1^ was achieved. This TOF value compares to other catalysts reported in the literature that have reached TOF values of 2184 h^−1^ at room temperature based also on Pd dispersion, although the actual dispersion percentage was not given [28]. 

It is important to note that gas phase analysis did not reveal detectable amounts of CO, methane or other products under the highest hydrogen production rates. Formation of CO from formate has been previously observed in other catalysts and it is derived from dehydration of formic acid. The presence of CO, even in minute proportions, is highly detrimental for H_2_ quality since CO is a strong poison of the noble metal catalysts used in fuel cells that must be coupled with on board H_2_ generation. In this context, the purity of hydrogen generated should be high enough to avoid damage of the fuel cell catalysts. The data with our Pd/CS-GO catalysts show that the hydrogen formed would be suitable for this application. In addition, it has also been observed that CH_4_ can be present in the reaction mixtures due to the occurrence of CO_2_ methanation promoted by the same catalyst activating formate decomposition. Thus, as the reaction progresses, H_2_ and CO_2_ accumulate in the gas phase, and the possibility of an unwanted reaction between these two gases increases. However, under none of the reaction conditions of the present study, formation of methane or light hydrocarbons was observed.

To justify the reasons why hydrogen generation is more efficient in methanol compared to water, some catalytic tests of NH_4_HCO_2_ decomposition in methanol adding at initial reaction time NaOH at a concentration below 10^−5^ M were performed. Coincident initial reaction rates of H_2_ generation as that of the run in the absence of any NaOH addition were measured, indicating that the presence of OH^-^ anion at the concentration corresponding to the experiments in aqueous solution (pH = 7) does not play any significant role in the lower activity of Pd/CS-GO2 deactivating the catalyst. Therefore, further studies are still necessary in order to understand the main factors responsible for the influence of the solvent on the catalytic activity of Pd/CS-GO2 for the decomposition of NH_4_HCO_2_.

As it can be seen in Figure 5, the temporal profile of H_2_ evolution shows that the reaction stops at about 125 min when the NH_4_HCO_2_ conversion was about 40%, decreasing significantly the reaction rate at longer reaction times and higher conversions. A possible explanation of these time-conversion plots would be that ammonia formed in the process according to Equation (1) deactivates Pd catalyst at high concentrations. In fact, the presence of ammonia was easily detected in the gas phase. This negative influence of ammonia at high concentrations was confirmed by performing a control experiment in which the reaction was started in the presence of a concentration of 10^−1^ M aqueous ammonia solution. It was observed that the initial reaction rate decreases significantly when ammonia is present in the medium. It is likely that due to the ability of NH_3_ to act as a Pd^2+^ ligand, this basic molecule competes with formate blocking gradually the active sites.

As commented for the reaction in water as solvent, also in the case of methanol, the evolution of CO_2_ was below stoichiometric amounts, a fact that is again explained by the solubility of CO_2_ in methanol. 

Catalyst stability was tested for Pd/CS-GO2, the material exhibiting the highest TOF value. It should be noted that Pd/CS-GO2 beads are apparently mechanically resistant and do not undergo breakage during the reaction. This mechanical robustness allows a better recovery of the catalyst just by quick sedimentation, making reusability tests easy to perform. Figure 10 shows the temporal profile of H_2_ evolution upon the reuse of Pd/CS-GO2 in methanol at 60 °C. A gradual deactivation of the catalyst occurs upon reuse. Chemical analysis of the methanol solution after filtration of the solid particles revealed that the percentage of Pd leached from the solid into the solution was negligible, ruling out Pd leaching as a deactivation pathway.

To determine if the cause of deactivation was agglomeration of Pd particle size as it was in the case of Pd/CS, TEM images of the reused catalyst were analyzed. It was observed that the average particle size of Pd NPs increased upon reuse from 1.7 nm for the fresh sample to 3.67 nm for the used Pd/CS-GO2, suggesting that this agglomeration of Pd NPs is the main cause of deactivation of the catalytic activity. It is proposed that changes in the CS-GO2 support during the reaction with a probable decrease of surface area and porosity is behind the observed Pd NPs agglomeration. This was the same reason as that proposed for the observed deactivation in Pd/CS that is attenuated by the presence of GO as cross-linker.

Thus, although the performance of Pd/CS-GO2 measured in TOF values is high, issues such as conditions for complete NH_4_HCO_2_ decomposition and increased catalyst stabiliity are yet to be solved in future studies. One of the main advantages of Pd/CS-GO2 is the easy scale-up of the catalyst, due to the availability of starting materials and the preparation procedure. 

## 3. Materials and Methods 

### 3.1. Synthesis of Pd/CS Aerogels and Pd/CSGO Aerogels

Chitosan MMW (1000 mg; ACROS) was dissolved in 50 mL of 1.25 vol% acetic acid solution. A complete dissolution was obtained after 21 h of stirring at room temperature. CS beads were formed by dropping the solution through a 0.8 mm syringe into 4 M NaOH solution. CS beads were kept in the alkaline solution for 3 h followed by extensive washings until washing water attained neutral pH. The hydrogels obtained were then immersed into a 19 mM Pd(NH_3_)_4_Cl_2_ × H_2_O solution for 24 h. The Pd containing alcogels were obtained by exchanging water with ethanol at increasing ethanol/water ratios (10/90; 30/70; 50/50; 70/30; 90/10; and 100% ethanol). Pd was reduced by immersing for 10 min the alcogels in a solution of 20 mg of NaBH_4_ dissolved in 100 mL of ethanol. Upon immersion in NaBH_4_ solution, the beads turned black immediately confirming the reduction of Pd.

A similar procedure was used to synthesize Pd/CS-GO beads, but prior to forming the chitosan beads, a certain amount of GO (corresponding to 3, 7, and 12 wt%) was sonicated in 50 mL of 1.25 vol% acetic acid solution for 90 min. The GO containing solution was added to 1000 mg of chitosan. Complete dissolution of chitosan was obtained after 21 h. The alcogels were obtained by the same procedure described above. Dry Pd/CS and Pd/CS-GO catalysts were obtained using an automated critical point dryer (Leica EM CPD300). Figure 11, Figure 12 and Figure 13 show EDX images of Pd/CS, Pd/CS-GO-2, and Pd/CS-GO-3.

### 3.2. Catalytic Activity Test

The experiments were conducted in a 130 mL reaction vessel under steady stirring (250 rpm) at the appropriate temperature ranging from 25 to 60 °C. The vessel was sealed and the internal air was degassed three times using argon. Typically, 50 mg of the catalyst (2.5 wt% of Pd) was added to a 10 mL of 1 M ammonium formate solution (10 mmol of ammonium formate). For comparison, the catalytic activity of a commercial 10% Pd/AC sample was also tested. The amount of Pd present in the reaction in the case of Pd/CS-GO2 used was 11.75 µmol in all experiments. H_2_ and CO_2_ generation was followed by analyzing the gases on the headspace using an Agilent 490 MicroGC having two channels both with TC detectors and Ar as the carrier gas. One channel has a MolSieve 5A column and analyses H_2_. The second channel has a Pore Plot Q column and analyzes CO, CH_4_ and light hydrocarbons. No evidence of the formation of CO (detectable in the equipment) was obtained. Quantification of the percentage of each gas was based on prior calibration of the system injecting mixtures with a known percentage of gases. Hydrogen generation rates were based on the amount of metal contained on the solid catalyst taking into account the metal dispersion value of 0.21 measured by CO adsorption. 

## 4. Conclusions

The present study has shown that Pd NPs of very small size adequate to promote formate decomposition can be obtained using large surface area CS as support. The presence of GO in an optimal proportion increases the catalyst’s activity and stability by enhancing the mechanical stability of the catalyst beads without increasing fragility. The formation of detectable amounts of CO or CH_4_ was not observed during formate decomposition. Methanol was found a suitable solvent for the reaction and a TOF value of 2279 h^−1^ was measured at 60 °C. This TOF value is among the highest ever reported for formate decomposition at moderate temperatures. It was observed that the Pd/CS-GO2 catalyst is deactivated upon reuse, the main reason being Pd agglomeration and particle growth due to the decrease in CS porosity and surface area. Further work is in progress to increase catalyst stability by increasing rigidity. Therefore, the present study constitutes a clear example showing the potential of natural chitosan as support for transition metal NPs to develop highly efficient catalysts.

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
