# Peer review of "Palladium Supported on Porous Chitosan–Graphene Oxide Aerogels as Highly Efficient Catalysts for Hydrogen Generation from Formate"

_molecules, 2019, doi:10.3390/molecules24183290_

Round 1

Reviewer 1 Report

Figs. 4/5/6/7 should include a legend with the different colors and symbols inside the plot. It is difficult to differentiate based only on the figure caption.

I would suggest to include supplementary material in the main text. The supplementary material is not that extended to define a separate document. Moreover, there seems to be good quality information in the supplementary material that needs to be highlighted.

Please highlight even more why ammonium formate decomposition reaction was selected? Can other substances be used with the same catalysts?

The paper should include a separate paragraph (perhaps before section 4?) that will summarize the results of this study. Questions such as a) impact of this study to research community, b) applicability of the process in a scaled-up system, c) advantages and disadvantages, d) future strategy for improvement and so on should be discussed. In this way, the actual novelty and importance of this study will be highlighted.

Author Response

Reviewer 1

Principio del formulario

Figs. 4/5/6/7 should include a legend with the different colors and symbols inside the plot. It is difficult to differentiate based only on the figure caption.

Figures have been edited as suggested.

I would suggest to include supplementary material in the main text. The supplementary material is not that extended to define a separate document. Moreover, there seems to be good quality information in the supplementary material that needs to be highlighted.

Following reviewer’s suggestion, the supporting information has now been merged into the main text.

Please highlight even more why ammonium formate decomposition reaction was selected? Can other substances be used with the same catalysts?

Revised version has now added another sentence in the last paragraph before the results section explaining the reasons why ammonium formate was selected as substrate. Other formates can also be used as liquid hydrogen carriers. 

The paper should include a separate paragraph (perhaps before section 4?) that will summarize the results of this study. Questions such as a) impact of this study to research community, b) applicability of the process in a scaled-up system, c) advantages and disadvantages, d) future strategy for improvement and so on should be discussed. In this way, the actual novelty and importance of this study will be highlighted.

We thank the reviewer for this comment. Most of the requested comments are in section 5 Conclusions. However, following the reviewer’s advice, a separate paragraph has now been added before section 4 summarizing the impact of the study, its applicability in scaled-up system, advantages and disadvantages and future strategies for improvement.

Reviewer 2 Report

The subject of the paper and the results obtained provide new insight into the process of hydrogen production by formate decomposition. 

The following comments should be taken into account in the revision of the manuscript:

Activation energy is measured in kJ mol-1, but not in kJ mol-1 K-1. The state of Pd after catalysis should be studied byXPS, in particular the Pd(0)/Pd(2+) ratio. In the best case of the Pd/CS-GO-3 catalyst, the conversion of formate to H2 does not exceed 40-50%. This is far below the quantitative conversion, which is required by the use of formate on board as a liquid hydrogen storage carrier. Ammonia is formed as a product of decomposition. Is there any influence of the ammonia on the stability and state of Pd?

Author Response

Reviewer 2

The subject of the paper and the results obtained provide new insight into the process of hydrogen production by formate decomposition. 

The following comments should be taken into account in the revision of the manuscript:

Activation energy is measured in kJ mol-1, but not in kJ mol-1 K-1.

We thank the reviewer for bringing this error to our attention.

The state of Pd after catalysis should be studied byXPS, in particular the Pd(0)/Pd(2+) ratio.

As indicated above Figure 3, XPS has been used to determine the Pd(0)/Pd2+ ratio of the exposed external layers of the particles. This point has been stressed in the revision.

In the best case of the Pd/CS-GO-3 catalyst, the conversion of formate to H2 does not exceed 40-50%. This is far below the quantitative conversion, which is required by the use of formate on board as a liquid hydrogen storage carrier. Ammonia is formed as a product of decomposition. Is there any influence of the ammonia on the stability and state of Pd?

The reviewer is correct. The presence of NH3 in the solution, decreases the reaction rate, meaning that it is a deactivating agent. The experiment in the presence of NH3 was commented three paragraphs before Figure 7. This paragraph has been now reinforcing adding a sentence that explicitly emphasized the negative role of NH3 in the ammonium formate decomposition.

Round 2

Reviewer 1 Report

No further change is required.

Reviewer 2 Report

The authors revised the manuscript and took into account all the comments